# RETHINKING THE OOD GENERALIZATION FOR DEEP NEURAL NETWORK: A FREQUENCY DOMAIN PERSPECTIVE

## ABSTRACT

Out-of-distribution (OoD) generalization has long been a challenging problem that remains largely unsolved. Despite numerous attempts to generalize image classification models to OoD datasets, few novel proposals have surpassed the classical Empirical Risk Minimization (ERM) methodology systematically. In this work, we revisit the frequency-based analysis of OoD generalization for images. Based on the Shapley value, a theoretical measure in game theory, we quantify the influence of each frequency component on the model's performance. With this analysis, we can explain the model's performance statistically. We observe that although the fallacious outputs of our model on OoD generalization tasks frequently stem from low-frequency components of OoD images, the interference pattern is highly class-wise. To further exploit our observation, we propose Class-wise Frequency Augmentation (CFA) to augment favorable frequency components and inhibit unfavorable ones. This approach can greatly improve the performance of existing OoD generalization algorithms. Our extensive experiments on five baseline OoD algorithms across seven OoD datasets provide encouraging results that prove the effectiveness of CFA on OoD generalization. Especially, CFA outperforms the state-of-the-art methods with the most substantial improvement on ColoredMNIST, increasing the identification accuracy from 60.2% to 73.0%.

## 1 INTRODUCTION

A plethora of deep learning algorithms have demonstrated their utility in practical applications (Affonso et al., 2017; Lopez & Kalita, 2017), but these algorithms heavily rely on the independent and identically distributed (i.i.d.) assumption. However, the assertion that the training dataset resembles the test dataset in data distribution is proved unreliable by multiple practical applications. Neural networks can be misguided by input shifts imperceptible to humans, leading to prominent performance drops in practice (Szegedy et al., 2014). This issue severely curbs the generalization effectiveness of the model, leading to the prominent Out-of-Distribution (OoD) generalization problem.

According to Ye et al. (2022), the OoD distribution shift can be divided into two patterns: diversity shift and correlation shift, referring to different reasons for domain shift. In practical situations, input data samples are often affected by both types of distribution shifts simultaneously. Few methods produce satisfactory outcomes for both shifts, and it has become commonplace that novel methods even fail to outperform the simple empirical minimization method (ERM, Vapnik (1999)). To learn more domain-generalizable features, some methods (Xu et al., 2021; Lin et al., 2023) are proposed to focus on frequency domain instead of pixel-wise analysis. Lin et al. (2023) proposes the Deep Frequency Filtering (DFF) modules to modulate the frequency components. This approach is capable of enhancing the transferable frequency components and suppressing the ones not conducive to generalization in the latent space.

In our work, we revisit the frequency-based analysis of OoD generalization from a new perspective. Different from previous work (Lin et al., 2023) modifying the network to modulate the frequency components, we resort to Shapley value (Shapley, 1953) to precisely attribute which frequency components of training images are important for generalization. Our strategy is to manipulate the

Figure 1: Illustration of calculating Shapley Value calculated in frequency domain and Shapley Value of different classes in frequency domain.

training data and thus create a *model-agnostic* method that can be applied to various networks to further boost their performance on OoD tasks.

As a theoretical result of the game theory, Shapley values provide feasible and general quantification for the contribution of different frequency components, by averaging the marginal contribution of each frequency component (Hu et al., 2022; Chen et al., 2022). As an example, Figure 1 illustrates the quantified contribution of frequency components to the output of the model. This measure enables the extraction of positive- and negative-contribution frequency components. We plot average Shapley values over frequency components of different class from different domains in Figure 1. We refer to the frequency components with positive contribution to classification as positive frequency components (PFC) and those with negative contribution to classification as negative frequency components (NFC). For the same class, no matter what domain it comes from, the positive frequency components are supposed to contain some domain-invariant features. If we aggregate these positive frequency components from different domains, this may help the network learn more domain-generalizable features. Besides, as can be seen in Figure 1, different classes exhibit largely different preferences of PFCs and NFCs. Take the class "giraffe" as an example, its PFCs mostly lie in low bands and NFCs lie in middle bands. Take the class "house" as a comparison, its PFCs mostly lie in low and middle bands while its NFCs lie in low bands, which reveals that the influence of different frequency components on the output of the model is largely class-wise. The distribution of Shapley value of images within a class bears certain resemblance, hinting the possibility to propose an algorithm utilizing this pattern.

Based on the analysis above, we propose a natural data augmentation strategy, Class-wise Frequency Augmentation (CFA) to improve the OoD performance. Our algorithm is devised to enhance the frequency components with positive contributions to the performance of the model and inhibit those with negative effects. Empirically, this algorithm could improve the OoD performance of the model significantly.

Our major contributions are as follows:

- We explore Shapley value, a theoretical measure in game theory, to quantify the contribution of each frequency component of images to the prediction results of the models in OoD generalization.
- Based on Shapley value, we empirically explain why the performance of some algorithms deteriorates rapidly in OoD environments and why some algorithms outperform others in one type of domain shift while fail in another one.
- According to the interpretation of the performance of algorithms on OoD datasets, we propose Class-wise Frequency Augmentation Algorithm (CFA) to enhance the generalization ability on OoD datasets, which can be seamlessly integrated into other OoD generalization algorithms. We conduct experiments to verify the effectiveness of our method on five baseline OoD algorithms on both diversity shift and correlation shift. With CFA, we achieved the state-of-the-art results on benchmark datasets.

## 2 PRELIMINARY

**Out-of-distribution generalization.** As shown in Figure 2, the OoD distribution shift is divided into two patterns: diversity shift and correlation shift(Ye et al., 2022). Diversity shift is caused by the appearance of new unseen environment features in the test set while correlation shift delineates the spurious correlation between environment features and labels. Based on this metric, the majority of OoD countermeasures can only outperform ERM in at most one of the two categories. In practical situations, input data samples are often affected by both types of distribution shifts simultaneously,

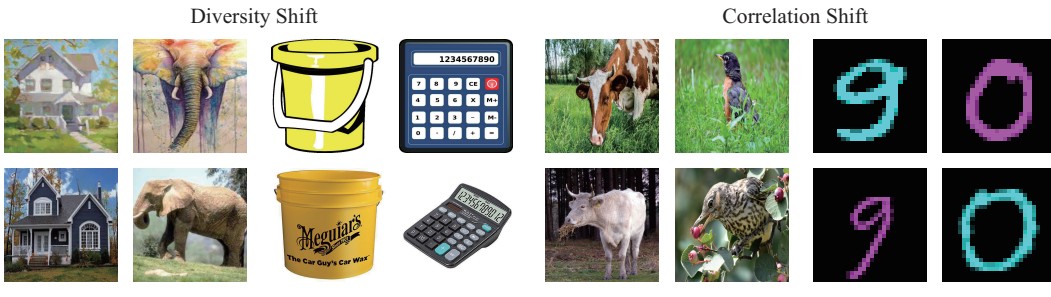

Figure 2: Examples of diversity shift and correlation shift. In practical scenarios, input samples are often affected by both types of distribution shifts simultaneously, while most algorithms can only outperform ERM in at most one of the distribution shifts. Thi motivates us to investigate more general OoD generalization methodologies.

leading to more complex spurious correlation. Among all the methods benchmarked, IRM(Invariant Risk Minimization,Arjovsky et al. (2020)) and RSC(Representation Self-Challenging,Huang et al. (2020)) exhibit strong preferences towards diversity shift or correlation shifts respectively but cannot perform well on the other type of shift while no statistical analysis has been performed to reveal the reason.

**Shapley value.** Shapley value is a theoretical result of the cooperative game theory (Shapley, 1953). The Shapley value of a player during a game is defined as the average of its marginal contribution over all conditions. This method relies solely on the input and output data of a model, which makes it a feasible way to evaluate the influence of each input factor on a blackbox model.

The definition of Shapley value is given as follows: $\mathcal{S} = \{1, \ldots, S\}$ denotes the $S$ factors jointly determining the output of a model. The performance of a specified input combination can be evaluated by a mapping $V : 2^S \to R$, which satisfies the property $V(\emptyset) = 0$. Let $\pi \in \Pi(\mathcal{S})$ be a specific permutation of set $\mathcal{S}$, indexing an element $i$ by $\pi(i)$. The set $P_i^\pi = \{j \in \mathcal{S} | \pi(j) < \pi(i)\}$ denotes the predecessors of a factor $i$. Finally, the Shapley value of a factor $i$ is given by:

$$\Phi(i) = \frac{1}{S!} \sum_{\pi \in \Pi(S)} [V(P_i^\pi \cup \{i\}) - V(P_i^\pi)] \tag{1}$$

Allowing for the existence of permutation in the calculation of Shapley value, a direct calculation seems computationally expensive. Fortunately, to reduce the computation, Castro et al. (2009) inspires us randomly sampling a portion of all permutations to evaluate the Shapley value,

$$\Phi(i) \approx \frac{1}{m} \sum_{\pi \in \Pi_m(\mathcal{S})} [V(P_i^\pi \cup \{i\}) - V(P_i^\pi)], \tag{2}$$

where $\Pi_m(\mathcal{S})$ denotes the $m$ permutations randomly chosen from the permutation set $\Pi(\mathcal{S})$.

**Discrete Fourier Transform.** As a classic signal processing technique, Discrete Fourier Transform (DFT) can convert finite signal sample series into discrete frequency components of the same length. For an image as a matrix $X \in R^{d_1 \times d_2}$, the 2D DFT is defined as follows,

$$F(X)(u, v) = \sum_{m=0}^{d_1-1} \sum_{n=0}^{d_2-1} X(m, n) e^{-2\pi i (\frac{mu}{d_1} + \frac{nv}{d_2})} \tag{3}$$

where $F$ denotes Discrete Fourier Transform; u and v denotes spatial coordinates ; m and n denotes frequency coordinates.

Correspondingly, its 2D inverse DFT is given as,

$$X(m, n) = \frac{1}{d_1 d_2} \sum_{u=0}^{d_1-1} \sum_{v=0}^{d_2-1} F(X)(u, v) e^{2\pi i (\frac{mu}{d_1} + \frac{nv}{d_2})} \tag{4}$$

## 3 FREQUENCY-BASED ANALYSIS FOR OoD GENERALIZATION

### 3.1 SHAPLEY-VALUE BASED FREQUENCY COMPONENT INFLUENCE QUANTIFICATION

With the assistance of Shapley value, we can quantify the influence of each frequency component on the output of the model. For each data sample $(X, y)$, we assume that $d_1 \times d_2$ frequency

components $F(X)(0,0), \ldots, F(X)(d_1 - 1, d_2 - 1)$ are the players in the scenario of a cooperative game. Therefore, the player set is given as:

$$S = \{F(X)(0,0), \ldots, F(X)(d_1 - 1, d_2 - 1)\} \qquad (5)$$

To assess the performance of a player combination, we resort to inverse DFT to transform the frequency combination back to original space, and utilize the output of the model as the assessment index. To be more specific, the function $V : 2^S \to R$ is instantiated as:

$$V(I[F(x) \in T] \otimes F(X)) = f(F^{-1}(I[F(x) \in T] \otimes F(X))) - f(\emptyset) \qquad (6)$$

where $\otimes$ denotes Hadamard product, $I$ is an indicator function that returns one if the element belongs to the designated set $T$, and $T \subset S$. $f$ means the output of model. For classification task, it means the probabilities. Next, the Shapley value of a particular frequency component can be calculated as:

$$\phi_{u,v}^X = \frac{1}{m} \sum_{\pi \in \Pi_m(S)} V(P_{u,v}^\pi \cup \{F(X)(u,v)\}) - V(P_{u,v}^\pi) \qquad (7)$$

in which $P_{u,v}^\pi$ denotes the predecessor set of $F(X)(u,v)$, i.e. $P_{u,v}^\pi = \{F(X)(u',v') \in S | \pi(u',v') < \pi(u,v)\}$ under the specific permutation $\pi \in \Pi(N)$. The $d_1 \times d_2$ Shapley values of an input image form a matrix, namely

$$\Phi(F(X)) = (\phi_{u,v}^X) \qquad (8)$$

This expression quantifies the impact of frequency components on the output of the model, which facilitates further analysis. According to OoD-Bench, we consider two typical datasets PACS and ColoredMNIST dominated by diversity shift and correlation shift respectively.

## 3.2 SHAPLEY-BASED ANALYSIS: DIVERSITY-SHIFTED DATASETS

In this part, we first conduct the Shapley-based frequency quantification on datasets dominated by diversity shifts. We take PACS as an typical example. We plot the Shapley value versus frequency components for different algorithms (including ERM, IRM and RSC) in Figure 3. Through analyzing the distribution of the Shapley value for ERM algorithm on correctly and wrongly classified images of PACS, we conclude that the positive Shapley value of the correctly classified ones mostly concentrate on the low-frequency bands, and is much higher than that in the high-frequency counterparts. In the frequency band ranging from 1 to 5, the Shapley values of correctly classified samples range from 0.3 to 0.6; while the value for the wrongly-classified samples possesses much lower intensity (less than 0.1) in these low frequency bands. Some are even less than zero and thus induce significantly negative effects for classification.

RSC outperforms ERM and IRM in datasets dominated by diversity shift with lower errors on OoD domains (Huang et al., 2020; Ye et al., 2022). In the previous paper Huang et al. (2020), the improved performance is attributed to that RSC discards the dominant features activated on the training data, and forces the network to activate remaining features that correlate with label. While this intuitive explanation is helpful, our Shapley-based observation provides new perspectives on the phenomenon: all the three approaches mainly rely on the low- and medium-frequency components to conduct the classification, while only RSC is able to filter out harmful components that result in wrong classification in low-frequency band. For ERM and IRM, the negative influence in low-frequency band leads to the overall degraded OoD generalization performance. For further illustration, we reconstruct some images from different frequencies. As can be seen in Figure 4 , the image reconstructed from positive frequency retains key information such as the outline of the object, which is helpful for classifying the object. This type of features is more domain-invariant and helps alleviate OoD problems. However, the image recovered from the negative frequency is blurred, mainly focus on color and irrelevant to the object category.

## 3.3 SHAPLEY-BASED ANALYSIS: CORRELATION-SHIFTED DATASETS

In this part, we analyze the performance of OoD algorithms on datasets dominated by correlation shift based on Shapley value quantification. ColoredMNIST is a widely-used dataset dominated by correlation shift in OoD generalization. This dataset caters to a binary classification task. In the training dataset, labels are strongly correlated with digits' color, while this false correlation is totally

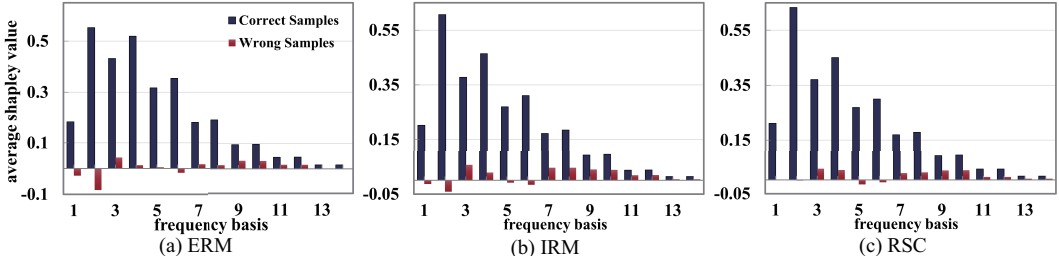

Figure 3: PACS dataset: Shapley values for different frequency components.

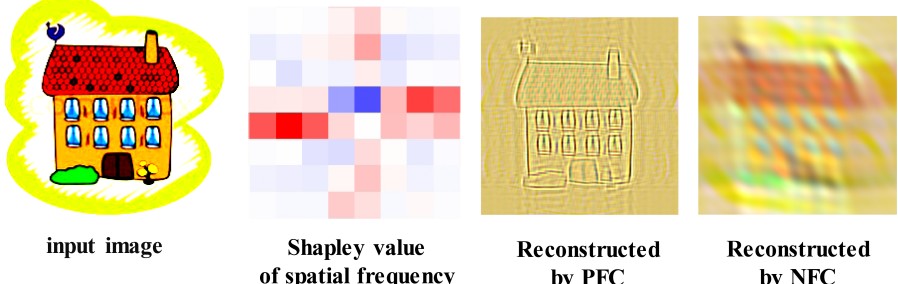

Figure 4: Image reconstructed with different frequency components from PACS dataset.

reversed in the test dataset. Leveraging the understanding of Shapley value on frequency components, we conduct a comprehensive analysis of the factors that contribute to various networks producing expected or erroneous judgments on input images, as shown in Figure 5. One major observation is that the network's tendency to produce incorrect judgments primarily arises from low-frequency factors in the images. Conversely, there appears to be a notable correlation between the abundance of mid-frequency information and the likelihood of the model generating the expected outcome. This phenomenon has inspired us to assist the neural network in effectively digesting mid-frequency information while curbing its inclination to focus on low-frequency information, which is obviously correlated with the color of input images in this situation.

Compared to ERM and RSC, IRM demonstrates a significant advantage in the classification task when dealing with datasets that exhibit correlation shift. As illustrated in Figure 5, IRM places more emphasis on middle-frequency components, which are generally independent of colors. Conversely, RSC and ERM yield unsatisfactory performance on ColoredMNIST, as they tend to focus on low-frequency components that are relatively misleading. While in the original IRM paper, the superior performance of IRM is interpreted as the result of learning invariant representations across training domains, our method provides a visually intuitive statistical method to explain this phenomenon.

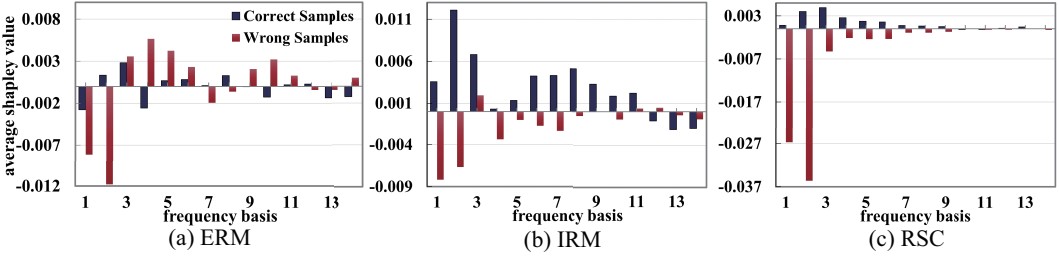

Figure 5: ColorMNIST dataset: Shapley values for different frequency components.

## 4 METHOD

Based on DFT and Shapley value, we now propose a feasible method that improves the performance of the model on OoD tasks. The definition of the Shapley value allows us to categorize frequency components based on the positivity of their Shapley value. An intuitive thought would be to exploit the information rendering satisfactory outcome as a modifier to intensify input features. It is expected that sample image after modification will possess additive common feature within the class, enriching the class information for the model to digest. Thus, this operation induces the model to focus more on common feature within the class, instead of the erroneous OoD environment correlation,

which, according to our experiment, boosts the performance of the model on OoD generalization prominently.

## 4.1 CLASS-WISE FREQUENCY AUGMENTATION

We start from $n$ data samples $\{(X_1, y), \ldots, (X_n, y)\}$ extracted from a specific class $y$ of various domains. To study the problem on the frequency domain, we first implement a DFT operation on all sample points, generating $\{F(X_1), \ldots, F(X_n)\}$, which enables further analysis via Shapley value in the frequency domain. Each sample $F(X_i)$ generates corresponding Shapley value matrix $\Phi(F(X_i)) = (\phi_{u,v}^{X_i}), \phi_{u,v}^{X_i} = \frac{1}{m} \sum_{\pi \in \Pi_m(N)} V(P_{u,v}^\pi \cup \{F(X_i)(u,v)\}) - V(P_{u,v}^\pi)$ that equals $F(X)$ in size. The positive and negative frequency components can be expressed as:

$$p_{X_i}^{\hat{f}} = I[\Phi(F(X_i)) > 0] \otimes F(X_i) \quad n_{X_i}^{\hat{f}} = I[\Phi(F(X_i)) < 0] \otimes F(X_i) \tag{9}$$

respectively, where $\otimes$ denotes Hadamard product and $I$ denotes an indicator operation. With $n$ data samples available, we intend to search for the desired frequency combination $p_r^{\hat{f}}$ and $n_r^{\hat{f}}$, with which we can modify the data to improve the learning effect. The intention is clarified as follows:

$$p_r^{\hat{f}}(u, v) = \text{argmax}_{p^{\hat{f}}} \sharp \{\Phi(p^{\hat{f}}) > 0\} \quad n_r^{\hat{f}}(u, v) = \text{argmax}_{n^{\hat{f}}} \sharp \{\Phi(n^{\hat{f}}) < 0\} \tag{10}$$

where $\sharp$ denotes the cardinal function. In the vector space, the $n$ data samples decides $n$ vectors $p_{X_i}^{\hat{f}}, i \in 1, \ldots, n$ with corresponding cardinal of positive Shapley value components $\sharp \{\Phi(F(X_i)) > 0\}$. Consider the properties of distribution of frequency energy of an image, it is appropriate to assume that the desired vector lies at the center of weight of the vector cluster. Following this thought, we can approximate $p_r^{\hat{f}}$ and $n_r^{\hat{f}}$ by $p_r^{\hat{f}} = \frac{1}{d_1 d_2} \sum_{i=1}^n p_{X_i}^{\hat{f}} \cdot \sharp \{\Phi(F(X_i)) > 0\}$ and $n_r^{\hat{f}} = \frac{1}{d_1 d_2} \sum_{i=1}^n n_{X_i}^{\hat{f}} \cdot \sharp \{\Phi(F(X_i)) < 0\}$ respectively.

Finally, the modification to input $X$ is implemented as follows:

$$F^{-1}(F(X) + \alpha p^{\hat{f}} - \beta n^{\hat{f}}) \tag{11}$$

where $\alpha$ and $\beta$ are both hyper-parameters. Based on the analysis above, we propose the specific algorithm 1 to boost OoD performance of the model. Figure 6 illustrates the process of our augmentation algorithm.

---

**Algorithm 1** Class-wise Frequency Augmentation for OoD generalization.

---

**Require:** Samples from train dataset from various domains: $\{(X_1, y), \ldots, (X_n, y)\}$.
**Ensure:** The model's parameters $\alpha$ and $\beta$.
 1: DFT transforms on input samples, generating $\{F(X_1), \ldots, F(X_n)\}$.
 2: Calculate the Shapley value matrix of each input sample via sampling m times: $\Phi(F(X_i)) = \frac{1}{m} \sum_{\pi \in \Pi_m(N)} V(P_{u,v}^\pi \cup \{F(X_i)(u,v)\}) - V(P_{u,v}^\pi)$.
 3: Extract the frequency components: $p_{X_i}^{\hat{f}} = I[\Phi(F(X_i)) > 0] \otimes F(X_i), n_{X_i}^{\hat{f}} = I[\Phi(F(X_i)) < 0] \otimes F(X_i)$.
 4: Calculate the possible frequency component which renders best and worst performance: $p_r^{\hat{f}} = \frac{1}{d_1 d_2} \sum_{i=1}^n p_{X_i}^{\hat{f}} \cdot \sharp \{\Phi(F(X_i)) > 0\}, n_r^{\hat{f}} = \frac{1}{d_1 d_2} \sum_{i=1}^n n_{X_i}^{\hat{f}} \cdot \sharp \{\Phi(F(X_i)) < 0\}$.
 5: Modify each train data sample $X_i$ by: $\hat{X}_i = F^{-1}(F(X_i) + \alpha * p_r^{\hat{f}} - \beta * n_r^{\hat{f}})$, which prepares intensified data to learn.
 6: Train the model with intensified data samples: $\theta = \text{argmin}_\theta E[l(f(\hat{X}_i), y)]$.

---

## 4.2 THEORETICAL ANALYSIS

In this section, we will provide theoretical analysis of CFA. We show that CFA can learn the ground-truth parameter more precisely than ERM.

We assume that input data $x$ can be decomposed into two components: $x_p$, and $x_n$, where $x_p$ denotes the causal frequency component consists of frequencies that contain semantic information about

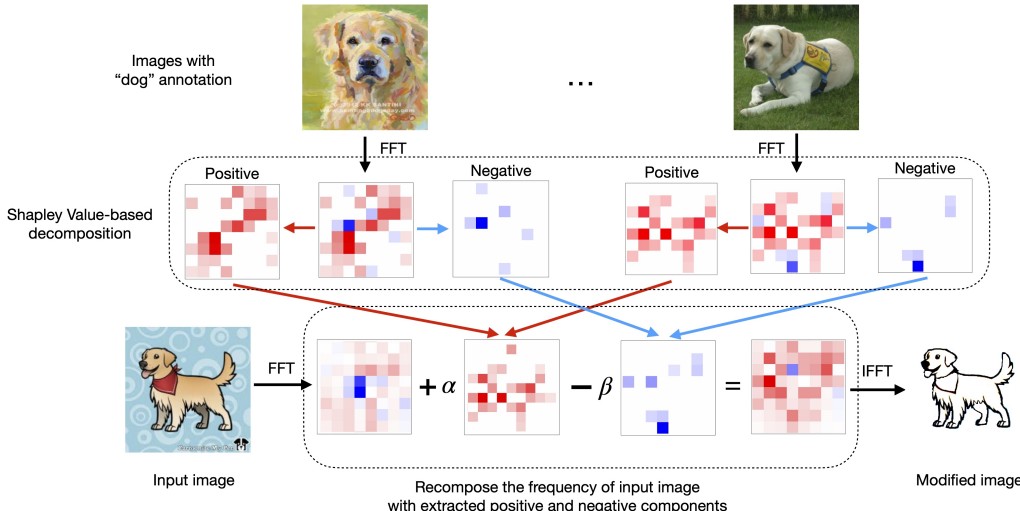

Figure 6: Process sketch of the Class-wise Frequency Augmentation algorithm. The "modified image" is an illustration used to emphasize that the category-related features becomes more prominent in the processed image.

the label $y$, and $x_n$ denotes the rest non-causal frequency components. The ground-truth model is assumed to be $y = (\tau^*)^T x_p + \epsilon$. Several mild assumptions on environmental noise $\epsilon$ are made, details of which are presented explicitly in the assumption in Appendix G. We first prove that the ground-truth model can be recovered if only the causal frequency component exist in the data:

**Theorem 1 (Parameter estimation without non-causal component, informal)** *If input data is free of NFCs, then the ground-truth model can be recovered without bias.*

Details of the proof can be found in Appendix G Theorem 3. This theorem inspires us to devise a method to optimize the estimation of model parameter via modifying PFCs and NFCs. Our algorithm is grounded on the following theorem:

**Theorem 2 (Provable lower error, informal)** *If PFCs and NFCs of inputs are modified according to CFA, strictly lower MSE error can be achieved in recovering the ground-truth causal model compared with ERM.*

The major contribution of our theory is to prove that CFA yields more accurate approximation of the ground-truth model. Detailed proofs of the theorem above can be found in Appendix G Theorem 5.

## 5 EXPERIMENTS

**Datasets and Settings.** To assess the effectiveness of our algorithm, we conduct comparison experiments with the ERM, IRM, RSC, CORAL (Sun & Saenko, 2016), W2D (Huang et al., 2022b) algorithms on the widely-used OoD datasets. For the datasets dominated by correlation shift, we experiment on Colored MNIST (Arjovsky et al., 2020), CelebA (Liu et al., 2015) and NICO (He et al., 2021). For the datasets dominated by diversity shift, we experiment on PACS (Li et al., 2017), OfficeHome (Venkateswara et al., 2017), and Terra Incognita (Beery et al., 2018).To further demonstrate the effectiveness of CFA, we also experiment on a large-scale dataset DomainNet (Peng et al., 2019). We use MLP as the backbone for the ColoredMNIST dataset and ResNet18 for other datasets. In order to enhance the performance of the state-of-the-art OoD algorithms with CFA, we design three training tactics: 1) Strengthen (S) the frequency components that yield a positive Shapley value within each class and domain. 2) Weaken (W) the frequency components that yield a negative Shapley value within each class and domain. 3) A combination of S and W. Further details on this training approach can be found in Appendix A.

### 5.1 ABLATION STUDY

We first implement ablation study on five OoD algorithms on the widely-used datasets on six OoD datasets. To make a strict ablation study, we run each experiment once with fixed random seed. The

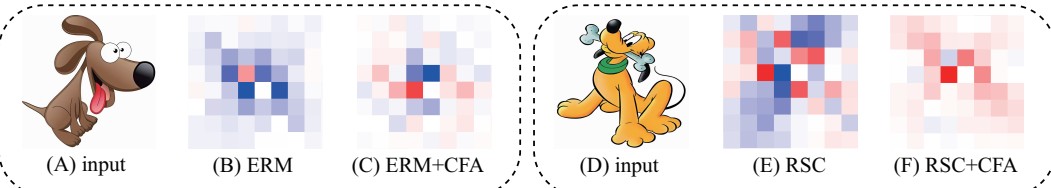

| (A) input | (B) ERM | (C) ERM+CFA | (D) input | (E) RSC | (F) RSC+CFA |

Figure 7: Heatmap of Shapley Value in frequency domain of misclassified samples with & without our CFA algorithm. Panel (B) and (E) are the heatmaps of misclassified samples of baseline algorithms, where the majority of frequency components exert negative influence on the classification mission. With CFA, revised components generally produce favorable contribution as shown in Panel (C) and (F), which produces expected classification results.

experimental results presented in Table 3 yield several key observations. Our experiments demonstrate that CFA consistently outperforms baseline algorithms across different domains. Specifically, the following findings can be highlighted: 1) The performance of the enhanced version of ERM is competitive with the baseline performance of RSC on PACS, validating the effectiveness of enhancing ERM through CFA. 2) When the S+W tactics are combined, CFA consistently outperforms five baseline OoD algorithms across six OoD datasets. This result indicates that the combined approach of CFA is actually the optimal choice for enhancing the performance of the baseline algorithms. CFA exhibits the most substantial improvement when combined with IRM, resulting in a significant increase in identification accuracy from 58.9% to 74.3% on ColoredMNIST,considering the fact that the maximum test accuracy can be achieved is 75% (Arjovsky et al., 2020). This also demonstrate that CFA is an universal mechanism can be seaminglessly incorporated into other OoD generealization algorithms to further improve their performances.

Table 1: Comparison of original algorithms and CFA-enhanced algorithms on different datasets. The Baseline column lists the results run by ourselves. The bold fonts indicate the best result. The red fonts indicate the improvements over baselines.

| Dataset | Alg. | Baseline/S/W/S+W | Dataset | Alg. | Baseline/S/W/S+W |
|---|---|---|---|---|---|
| Colored MNIST | ERM | 31.3 /33.5 / 34.8/ **35.0** (↑3.7) | PACS | ERM | 82.6 / 84.0/ 83.8/ **85.0** (↑2.4) |
| | IRM | 58.9 / 64.4 / 71.4/ **74.3** (↑15.4) | | IRM | 83.3 / 84.8 / 85.1/ **85.5** (↑2.2) |
| | RSC | 28.9 / 32.9 / 33.9/ **34.2** (↑5.3) | | RSC | 83.2 / 85.0/ 85.4/ **86.0** (↑2.8) |
| | CORAL | 31.5 / 34.9/ 33.2/ **36.0** (↑4.5) | | CORAL | 81.7 / 84.1 / 84.25/ **85.0** (↑3.3) |
| | W2D | 30.6 / 32.4 / 30.7/ **33.5** (↑2.9) | | W2D | 83.0 / 84.7 / 84.1/ **85.0** (↑2.0) |
| NICO | ERM | 73.8 / 74.0 / 76.0/ **76.0** (↑2.2) | Office Home | ERM | 61.7 / 62.7 / 62.8 / **63.1** (↑1.4) |
| | IRM | 71.2 / 72.0 /76.5/ **77.5** (↑6.3) | | IRM | 62.6 / 63.2 / 63.2 / **63.5** (↑0.9) |
| | RSC | 72.2 / 74.8/ 74.8/ **74.8** (↑2.6) | | RSC | 62.3 / 63.0 / 63.0 / **63.1** (↑0.8) |
| | CORAL | 70.2 / 76.5 / 73.8/ **77.2** (↑7.0) | | CORAL | 63.5 / 63.6 / 64.0 / **64.1** (↑0.6) |
| | W2D | 75.6 / 80.3 /79.7/ **81.1** (↑5.5) | | W2D | 63.3 / 64.1 / 64.1 / **64.2** (↑0.9) |
| CelebA | ERM | 87.6 / 88.0 / 88.8/ **89.0**(↑1.4) | TerraInc | ERM | 45.0 / 49.0 / 47.8 / **50.5** (↑5.5) |
| | IRM | 86.7 / 87.1 / 87.0/ **87.1** (↑0.4) | | IRM | 47.9 / 50.8 / 49.9 / **51.5** (↑3.6) |
| | RSC | 87.6 / 88.6 / 89.1/ **89.3** (↑1.7) | | RSC | 47.7 / 50.1 / 49.7 / **51.3** (↑3.6) |
| | CORAL | 87.1 / 87.9/ 87.7/ **88.1** (↑1.0) | | CORAL | 42.1 / 45.3 / 45.0 / **45.9** (↑3.8) |
| | W2D | 84.9 / 85.7 / 85.4/ **85.9** (↑1.0) | | W2D | 43.5 / 44.8 / 45.6 / **47.7** (↑4.2) |

To further illustrate the application of CFA, we visualize the heatmaps of Shapley value in the frequency domain for images misclassified by the baseline algorithms. As depicted in Fig 7, it can be observed that a significant portion of frequency components of misclassified images yield negative Shapley values. In contrast, when CFA is combined with ERM and RSC, respectively, the modified images exhibit additional positive Shapley values in the frequency components. Consequently, these positive Shapley values guide the respective algorithms towards correct classification.

## 5.2 COMPARISON WITH THE STATE-OF-THE-ARTS

The comparison between CFA and recently published state-of-the-art methods is presented in Table 2. We run each experiment with random seed for three times independently and report the average of three runs, as well as standard error.To further demonstrate the effectiveness of CFA, we also experiment on a large-scale dataset DomainNet. For DomainNet, we simply extend ERM with CFA to demonstrate its effectiveness in improving OoD generalization abilities on large-scale datasets. The table highlights that although all baseline methods exhibit explicit preferences towards either one of the two distribution shifts, the novel CFA stably outperforms baseline methods, irrespective of the inherent distribution shifts in the datasets. Notably, CFA achieves a new state-of-the-art

performance on six commonly-used datasets. Specifically, CFA successfully mitigates the distribution shift preferences of IRM and even outperforms some recently published strong baselines (Huang et al., 2022a). This addresses the challenge raised in the Ye et al. (2022) paper regarding the effectiveness of a method in dealing with both categories of distribution shifts simultaneously. Besides, in contrast to a recently proposed method DFF, which designs multiple filtering frequency modules to modulate the frequency components, our CFA is a training strategy in the data without any extra network design and can be directly applied to any model architecture. As it shown, our CFA shows great superiority over DFF on six datasets.

Table 2: Comparison with recent state-of-the-art methods. Among these methods, DFF modifies ResNet18 by replacing each convolution layer with filtering frequency modules, while our CFA is a training strategy without extra network design and directly trains on ResNet18. Results with † of DFF are provided in the paper and others are run by ourselves. It can be found that the proposed method is effective for both shifts types and achieve the best or comparable to best performances on all datasets.

| Algorithms | ColoredMNIST | CelebA | NICO | PACS | OfficeHome | TerraInc | DomainNet |
|---|---|---|---|---|---|---|---|
| ERM | 29.9 ± 0.9 | 87.2 ± 0.6 | 72.1 ± 1.6 | 81.5 ± 0.0 | 63.3 ± 0.2 | 47.7 ± 0.7 | 34.9 ± 0.5 |
| IRM | 60.2 ± 2.4 | 85.4 ± 1.2 | 73.3 ± 2.1 | 81.1 ± 0.3 | 63.0 ± 0.0 | 46.8 ± 0.4 | 25.0 ± 3.6 |
| RSC | 28.6 ± 1.5 | 85.9 ± 0.2 | 74.3 ± 1.9 | 82.8 ± 0.4 | 62.9 ± 0.4 | 48.0 ± 0.4 | 32.2 ± 0.5 |
| CORAL | 30.0 ± 0.5 | 86.3 ± 0.5 | 70.8 ± 1.0 | 81.6 ± 0.6 | 63.8 ± 0.3 | 43.8 ± 0.4 | 34.0 ± 0.4 |
| W2D | 31.3 ± 0.3 | 87.7 ± 0.4 | 71.9 ± 1.2 | 83.4 ± 0.3 | 63.5 ± 0.1 | 44.5 ± 0.5 | 28.0 ± 0.3 |
| DFF | 27.0 ± 0.4 | 84.5 ± 0.1 | 63.8 ± 2.1 | 82.2† | **67.6**† | 36.2 ± 0.3 | 25.4 ± 0.1 |
| CFA(ours) | **73.0 ± 0.7** | **88.3 ± 0.4** | **80.4 ± 0.3** | **84.6 ± 0.6** | 64.1 ± 0.0 | **50.7± 0.4** | **36.2 ± 0.1** |

## 6 RELATED WORK

**OoD Generalization** The concept of domain generalization was first introduced by Blanchard et al. Blanchard et al. (2011), in response to data distribution fluctuations in real-world applications. Over the years, various techniques have been proposed to enhance the out-of-distribution (OoD) generalization performance of models, which can be broadly categorized the following categories: 1) Domain alignment, a methodology that extracts domain-invariant features by aligning marginal distributions of training datasets. 2) Meta-learning Finn et al. (2017), a methodology that trains models using data from related tasks to learn domain shift while training to improve OoD generalization. 3) Data augmentation Goodfellow et al. (2016), a method that augments input data via data transformation Shi et al. (2020); Volpi & Murino (2019). Besides, some methods are proposed to focus on frequency domain instead of pixel-wise analysis. Xu et al. (2021) develop a novel Fourier-based data augmentation strategy called amplitude mix which linearly interpolates between the amplitude spectrums of two images to force the model to capture phase information. Besides, some theoretical studies have uncovered that CNNs have preferences for some frequency components. Wang et al. (2020) investigate the relationship between the frequency spectrum of image data and the generalization behavior of convolutional neural networks (CNN) and proposes that CNN can exploit high-frequency image components that are not perceivable to humans. Lin et al. (2023) propose Deep Frequency Filtering (DFF) for learning domain-generalizable features which is the first endeavor to explicitly modulate the frequency components of different transfer difficulties across domains in the latent space during training.

## 7 CONCLUSION

In this work, we have proposed the Shapley value-based analysis to quantify the contribution of different frequency components for the existing OoD approaches. This analysis have provided new perspectives on interpreting how these OoD generalization methods work across different types of datasets for image classification. Based on the obtained insights, we have developed a simple yet effective strategy to enhance the generalization performance of existing methods. CFA consistently outperforms five baseline OoD algorithms across seven OoD datasets.

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

## A    DETAILED SETTINGS FOR EXPERIMENTS

Our implementation of experiments is based on the OoD-bench(Ye et al., 2022) and DomainBed (Gulrajani & Lopez-Paz, 2021). For ColoredMNIST, we use a three-layer multi-layer perceptron (MLP) as the backbone. We first run a reasonable baseline for each method. Then we search $\alpha$ and $\beta$ as mentioned above. For other datasets, we use ResNet18 as the backbone. Our implementation is based on the OoD-bench suit. We first re-run these methods to check the implementations. Based on this setting, we incorporate our CFA in existing OoD methods to check whether CFA can universally imporve existing OoD generalization methods' performances. In CFA, we have two main hyperparameters: $\alpha$ is the degree to strengthen positive frequency bands and $\beta$ is the degree to weaken negative frequency. We first search $\alpha$ from (0.1-0.9,1.0-10.0) in S mode. Then we search $\beta$ from (0.1-0.9,1.0-10.0) in W mode.

When calculating Shapley values, for image resolution of 224, we divide the image to $16 \times 16$ patches in the frequency domain and sample $m = 1000$ times for each sample. For image resolution of 14, we divide the image to $14 \times 14$ patches in the frequency domain and sample $m = 1000$ times for each sample.

## B    DATASETS

1. Datasets dominated by diversity shift:

**PACS** contain images of objects and creatures depicted in different styles, which are grouped into four domains, photos, art, cartoons, sketches. In total, it consists of 9, 991 examples of dimension (3, 224, 224) and 7 classes.

**OfficeHome** has four domains: art, clipart, product, real,containing 15, 588 examples of dimension (3, 224, 224) and 65 classes.

**Terra Incognita** contains photographs of wild animals taken by camera traps at different locations in nature. We utilize fourlocations, L100, L38, L43, L46, covering 24, 788 examples of dimension (3, 224, 224) and 10 classes.

2. Datasets dominated by correlation shift:

**Colored MNIST** is a variant of the MNIST handwritten digit classification dataset. Following IRM(Arjovsky et al., 2020), this dataset contains 60, 000 examples of dimension (2, 14, 14) and 2 classes.

**NICO** consisting of real-world photos of animals and vehicles captured in a wide range of contexts such as "in water", "on snow" and "flying". There are 9 or 10 different contexts for each class of animal and vehicle. This dataset simulates a scenario where animals and vehicles are spuriously correlated with different contexts.

**CelebA** is a large-scale face attributes dataset with more than 200K celebrity images, each with 40 attribute annotations.We treats "hair color" as the classification target and "gender" as the spurious attribute.

3. Large-scale dataset:

**DomainNet** is a dataset of common objects in six different domains and 345 categories (classes). Domainet has around 600,000 images, while ImageNet-R has 200 ImageNet classes resulting in 30,000 images.

## C    AVERAGE SHAPLEY VALUE OVER FREQUENCIES OF EACH CLASS

In this section, we will demonstrate the average Shapley values for frequency components for each class. We plot average Shapley values over frequency components of each class for different algorithms including ERM(Vapnik, 1999) and RSC(Huang et al., 2020) in Figure 8. As can be seen, the blue bars represent the positive frequency components, and the red ones represent the negative frequency components. We refer to the frequency components with positive contribution to prediction as positive frequency components (PFC) and frequency components with negative

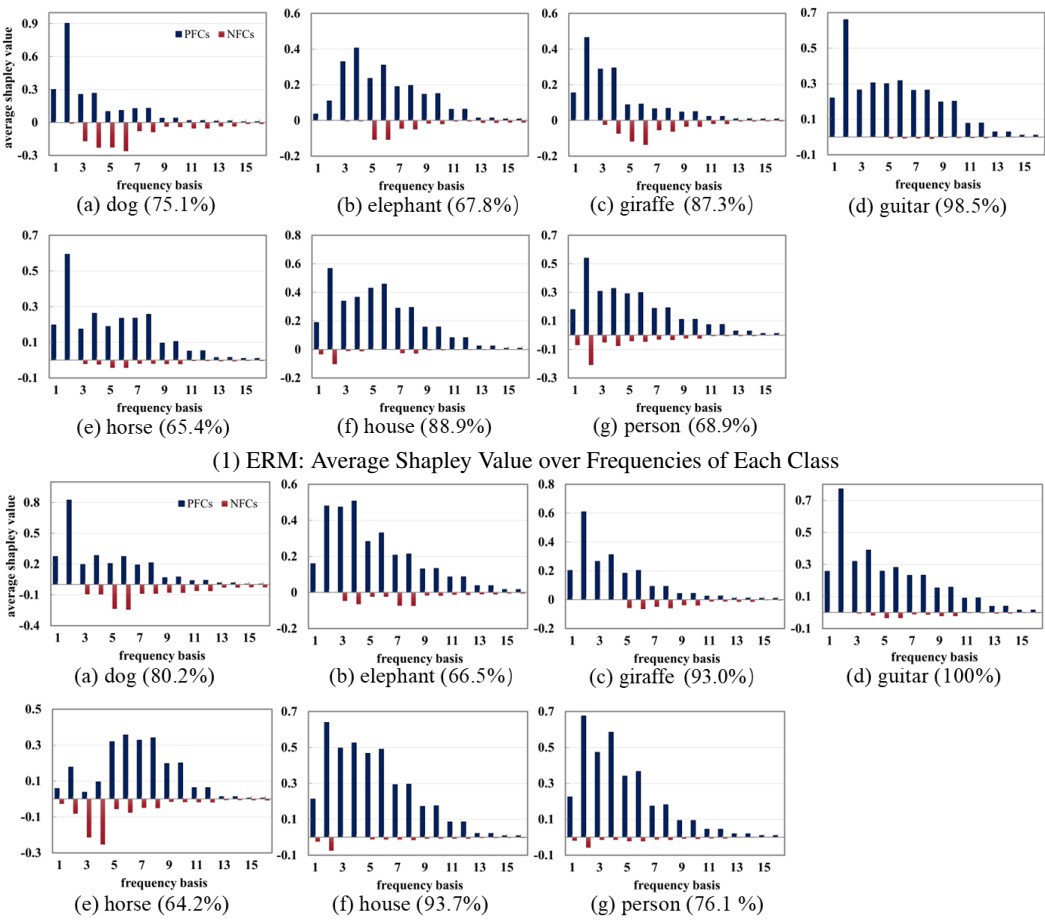

(1) ERM: Average Shapley Value over Frequencies of Each Class

(2) RSC: Average Shapley Value over Frequencies of Each Class

Figure 8: Average Shapley Value over Frequencies of Each Class for Different Algorithms.

contribution to prediction as negative frequency components (NFC). The percentage in brackets indicates the classification accuracy. As can be seen in Figure 8, different classes exhibit largely different preferences of PFCs and NFCs. Take the class "giraffe" as an example, its PFCs mostly lie in low bands and NFCs lie in middle bands. Take the class "house" as a comparison, its PFCs mostly lie in low and middle bands while its NFCs lie in low bands. This motivates us to consider strengthen or weaken specific PFCs and NFCs to improve OoD generalization performances. Besides, we can observe that the improvements of RSC over ERM may contribute the strengthen or elimination of specific frequency components. For example, in the histogram ERM(a) and RSC(a) in Figure 8 for the "dog" class, for ERM, the net effects are to focus on the lower frequency components, which leads to overfitting and degenerated performances, while for RSC, the net effects enables the DNN to consider multiple frequency components and avoid over-fitting. Take the class "person" as another example. From the histogram ERM(g) and RSC(g) in Figure 8, we can find that RSC weakens most of the NFCs and this leads to improved OoD performances. Therefore, based on above observations, we conclude that weakening the NFCs and strengthening necessary PFCs for each class can potentially can lead to improved OoD generalization performances and motivate the proposed algorithm.

# D    RESULTS OF OTHER NEURAL NETWORK ARCHITECTURE

We implement more experiments with ViT (small version) on two datasets. As it can be seen, CFA stills have impressive performance.

Table 3: Comparison of original algorithms and CFA-enhanced algorithms. The Baseline column lists the results run by ourselves. The bold fonts indicate the best result. We also list the improvements over baselines in the brackets.

| Backbone | Dataset | Algorithm | Baseline | S | W | S+W |
|----------|---------|-----------|----------|------|------|------------|
| ViT | CelebA | ERM | 82.0 | 84.4 | 83.7 | **85.0** (↑3.0) |
| | PACS | ERM | 82.4 | 86.0 | 85.8 | **86.7** (↑4.3) |

# E   PARAMETER SENSITIVITY ANALYSIS

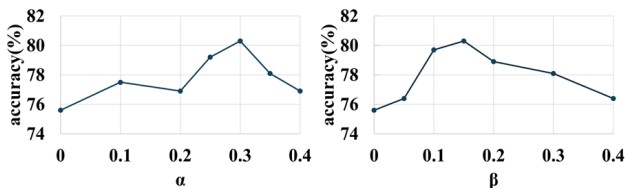

Figure 9: Parameter sensitivity analysis (W2D+CFA on OfficeHome).

In CFA, we have two main hyper-parameters: $\alpha$ is the degree to strengthen positive frequency bands and $\beta$ is the degree to weaken negative frequency. We evaluate the parameter sensitivity of CFA by changing one parameter and fixing the other. U-shaped parameter sensitivity curves are shown in Figure 9. From this figure, we can observe that suitable values of $\alpha$ and $\beta$ help to improve the generalization performances.

# F   THE LITERATURE OF EXPLAINABILITY WITH SHAPLEY VALUE

The Shapley value Shapley (1953) was initially proposed as a method for fairly distributing gains among a player coalition during a cooperative game based on their contributions. Its ability to quantify the contribution of each input factor makes it a desirable methodology for studying the explainability of neural networks Hu et al. (2022); Chen et al. (2022; 2018). To streamline this method, various techniques have been proposed. Some works Chau et al. (2022); Ancona et al. (2019) focus on accelerating its calculation through approximations, while others Frye et al. (2020a;b) investigate its application conditions. The ideal properties of the Shapley value, such as null player, efficiency, symmetry, and linearity, are pivotal to constructing more sophisticated algorithms.

# G   PROOF FOR RETHINKING THE OOD GENERALIZATION FOR DEEP NEURAL NETWORK

For simplicity of evaluation, we assume that input data $x$ consists of two components: $x = x_p + x_n$, where $x_p$ denotes the component that contains the frequencies rendering causal correlation with label $y$, while $x_n$ represents the non-causal part of $x$ which might lead to interference. To be more specific, the decomposition of x on frequency basis is $x = \sum_i x_i$.

We consider a simplified setting, linear relationship between $x_p$ and $y$. The regression model under assumption is given as follows.

**Assumption 1** *We assume the linear relationship between $x_p$ and $y$,*

$$y = (\tau^*)^T x_p + \epsilon, \tag{12}$$

*where $\epsilon$ is the noise residual. Formally, we further assume the noise residual $\epsilon$ follows the conditions,*

$$Cov(x_p, \epsilon) = \mathbf{E}[x_p] = 0. \tag{13}$$

Given that input data consist of nothing but causal signal for evaluation, i.e. $x = x_p$, it is viable to evaluate the ground-truth parameter of the label as follows:

**Theorem 3** *If input data $x \in \mathbb{R}^n$ satisfies $x = x_p$, the optimal estimation of $\tau$ in $y = (\tau^*)^T x + \epsilon$ is $\tau_P = \tau^*$.*

*Proof.* According to least square method, the optimal linear predictor for $y = (\tau^*)^T x + \epsilon$ is

$$\tau_P = (\mathbf{E}[x_p x_p^T])^{-1} \mathbf{E}[x_p y] \tag{14}$$

$$= \tau^* + (\mathbf{E}[x_p x_p^T])^{-1} (\mathbf{E}[x_p y] - \mathbf{E}[x_p x_p^T] \tau^*) \tag{15}$$

$$= \tau^* + (\mathbf{E}[x_p x_p^T])^{-1} \mathbf{E}[x_p (y - (\tau^*)^T x_p)] \tag{16}$$

$$= \tau^* + (\mathbf{E}[x_p x_p^T])^{-1} \mathbf{E}[x_p \epsilon] \tag{17}$$

$$= \tau^* + (\mathbf{E}[x_p x_p^T])^{-1} (\mathbf{E}[x_p] \mathbf{E}[\epsilon] + Cov(x_p, \epsilon)) \tag{18}$$

$$= \tau^* \tag{19}$$

In actual application, it is inevitable that there exists certain discrepancy between $x$ and $x_p$, i.e. $x = x_p + x_n$, where positive and negative frequency residual signal induces error to model parameters. The following theorem reveals the impact of spurious correlation between input data and environmental elements on the learning results of the model.

**Theorem 4** *If input data $x \in \mathbb{R}^n$ satisfies $x = x_p + x_n$, the optimal estimation of $\tau$ in $y = (\tau^*)^T x + \epsilon$ is $\tau_0 = \tau^* + (\mathbf{E}[x_p x_p^T])^{-1} \mathbf{E}[x_p \epsilon | x_n]$*

*Proof.*

$$\tau_0 = (\mathbf{E}[x_p x_p^T])^{-1} \mathbf{E}[x_p y] \tag{20}$$

$$= \tau^* + (\mathbf{E}[x_p x_p^T])^{-1} (\mathbf{E}[x_p y] - \mathbf{E}[x_p x_p^T] \tau^*) \tag{21}$$

$$= \tau^* + (\mathbf{E}[x_p x_p^T])^{-1} \mathbf{E}[x_p (y - (\tau^*)^T x_p)] \tag{22}$$

$$= \tau^* + (\mathbf{E}[x_p x_p^T])^{-1} \mathbf{E}[x_p \epsilon | x_n] \tag{23}$$

To illustrate the effectiveness of our data modification, we calculate the result of parameter assumption after data modification. The following theorem illustrates that modified data leads to $\tau'$, which is closer to model parameter $\tau^*$ than ordinary $\tau$ by norm.

**Theorem 5** *$\tau_S$, generated by fitting on the CFA-modified data, renders more accurate approximation of $\tau^*$ than $\tau$, i.e. $||\tau_S - \tau^*|| \leq ||\tau - \tau^*||$, where $\tau_S = \tau^* + (\mathbf{E}[(x_p + \alpha\hat{p})(x_p + \alpha\hat{p})^T])^{-1} \mathbf{E}[(x_p + \alpha\hat{p})\epsilon | x_n - \beta\hat{n}]$, and $\tau = \tau^* + (\mathbf{E}[x_p x_p^T])^{-1} \mathbf{E}[x_p \epsilon | x_n]$. $\hat{p} = \sum\limits_{\phi(x_i)>0} x_i, \hat{n} = \sum\limits_{\phi(x_i)<0} x_i$.*

*Proof.*

$$||\tau_S - \tau^*|| = ||(\mathbf{E}[(x_p + \alpha\hat{p})(x_p + \alpha\hat{p})^T])^{-1} \mathbf{E}[(x_p + \alpha\hat{p})\epsilon | x_n - \beta\hat{n}]|| \tag{24}$$

$$= ||(\mathbf{E}[x_p x_p^T] + \alpha\hat{p}(\alpha\hat{p})^T)^{-1} \mathbf{E}[(x_p + \alpha\hat{p})\epsilon | x_n - \beta\hat{n}]|| \tag{25}$$

According to eigenvalue perturbation theoryLöwdin (1962), Suppose $(\mu_i, \phi_i)_{i=0}^{\infty}$ are eigenvalue and unit eigenvector pairs of $\mathbf{E}[x_p x_p^T]$, i.e. $\mathbf{E}[x_p x_p^T] \phi_i = \mu_i \phi_i$ The following result can be deducted:

$$[\mathbf{E}[x_p x_p^T] + \alpha\hat{p}(\alpha\hat{p})^T] \phi_i' \tag{26}$$

$$= (\mu_i + \phi_i^T \alpha\hat{p}(\alpha\hat{p})^T \phi_i) \phi_i' \tag{27}$$

$$= (\mu_i + ((\alpha\hat{p})^T \phi_i)^T (\alpha\hat{p})^T \phi_i) \phi_i' \tag{28}$$

It is obvious that,

$$(\mu_i + ((\alpha\hat{p})^T \phi_i)^T (\alpha\hat{p})^T \phi_i) \geq \mu_i \tag{29}$$

and,

$$\alpha\hat{p}\mathbf{E}[(x_p + \alpha\hat{p})\epsilon | x_n - \beta\hat{n}] \leq \mathbf{E}[(x_p + \alpha\hat{p})\epsilon | x_n] \tag{30}$$

Therefore,

$$||\tau_S - \tau^*|| = ||(\mathbf{E}[(x_p + \alpha\hat{p})(x_p + \alpha\hat{p})^T])^{-1} \mathbf{E}[(x_p + \alpha\hat{p})\epsilon | x_n - \beta\hat{n}]|| \tag{31}$$

$$||(\mathbf{E}[x_p x_p^T] + \alpha\hat{p}(\alpha\hat{p})^T)^{-1} \alpha\hat{p}\mathbf{E}[\epsilon | x_n - \beta\hat{n}]|| \tag{32}$$

$$\leq ||(\mathbf{E}[x_p x_p^T])^{-1} \alpha\hat{p}\mathbf{E}[\epsilon | x_n - \beta\hat{n}]|| \tag{33}$$

$$\leq ||(\mathbf{E}[x_p x_p^T])^{-1} \mathbf{E}[\epsilon | x_n]|| = ||\tau - \tau^*|| \tag{34}$$

