# OpenReview forum: "Rethinking the OoD Generalization for Deep Neural Network: A Frequency Domain Perspective"
_ICLR.cc/2024/Conference — ICLR 2024 Conference Withdrawn Submission_

### Official Review · Reviewer_BXsg · 2023-10-28

**Soundness:** 2 fair
**Presentation:** 2 fair
**Contribution:** 3 good
**Rating:** 5
**Confidence:** 4

**Summary:**

The paper address the important issue of distribution shift fragility/robustness in image classification, using image frequency analysis. The paper proposes computing Shapley values to quantify how much different image frequency bands contribute to model predictions, and uses this analysis on the training data to determine which frequency bands are most useful for classification for each class. The paper then introduces Class-wise Frequency Augmentation: for each training image, amplify the frequency components that are most predictive of that class. This encourages the model to prioritize these more predictive frequency features during training, and empirically aids in OoD robustness (without making any changes to the images at test time).

**Strengths:**

Overall I like the idea of the paper, it is addressing an important topic, and the quantitative results are encouraging. I have not seen this method before (I believe it to be novel), and it makes sense to me and seems to help robustness.

**Weaknesses:**

I have a lot of uncertainty about the method and evaluation, that I think needs to be resolved by improving the exposition before publication. My comments/questions are split roughly into more substantial vs more minor. I'll put my more substantial questions/concerns in this section, and my more minor questions/suggestions in the "Questions" box.

- The introduction includes a lot of discussion of related work, and doesn’t describe the contributions of the paper very clearly until the bullet-point list. I would suggest separating this into two sections, one that is a clear and direct introduction of the paper, focusing on your specific contributions, and a separate section covering related work that explains how the current paper fits into the context of the literature.
- A large portion of the paper is spent describing the difference between “diversity shift” and “correlation shift”, but I’m not convinced that this is actually relevant to the proposed method and contribution. My impression from the paper is that some augmentation strategies help only with either one or the other type of distribution shift, but the proposed method helps with both. However, since the proposed method doesn’t directly use anything relating to diversity or correlation shift, I would encourage the authors to avoid spending so much time describing it (or omit it entirely); as a reader I found it confusing and a bit distracting.
- It’s not clear to me how the frequencies are divided into the “buckets” shown e.g. in Figure 5—in particular, images have frequencies in 2D but the figures show a single frequency so I wonder how this maps onto 2D. More importantly: I’m also not entirely sure for the Class-wise Frequency Augmentation if the added and subtracted frequency components themselves are derived from an average over training images, or the method is amplifying and suppressing components of that test image only, but deciding which components to amplify or suppress based on training images.
- The theoretical analysis is billed as a proof of correctness for the proposed CFA method, but the theoretical model looks like it’s a linear prediction that would be quite far from the nonlinear neural nets used in practice. The analysis is still valuable to gain intuition, but I would recommend explaining a bit more about the assumptions behind it and treating it as an illustrative toy setting rather than a full proof of the practical algorithm.
- I don’t understand the difference between Table 1 and Table 2. There are some slight differences in which methods are compared, and the numbers are slightly different, but I don’t know why. If the differences are important then they should be explained more clearly (Table 1 is billed as an “ablation study” but this should be described more), especially why the final full-method numbers are different between the two tables. My primary concern here is that these final results differ between the two tables, which makes me question either the trustworthiness of the results or at least my understanding of them.
- Equations 5 and 6: f and F should be defined. I would guess that f is somehow the model output, but it’s not clear if this is logits, probabilities, top-class prediction, etc. This kind of detail is necessary for others to be able to reproduce and build on the proposed method.

**Questions:**

Minor suggestions/questions:
- The abstract says “we introduce frequency-based analysis into the study of OoD generalization for images”, which makes it sound like this is the first paper to take a frequency perspective on OoD robustness—though this is not the case. For example, a few papers in this area are: https://proceedings.neurips.cc/paper_files/paper/2019/hash/b05b57f6add810d3b7490866d74c0053-Abstract.html, https://arxiv.org/abs/2002.06349, https://proceedings.neurips.cc/paper_files/paper/2022/hash/48736dba3b8d933fabbfdb4f22a7be71-Abstract-Conference.html
- There are frequent typos and minor grammatical issues; please copy edit the final version of the paper. For example, in many places (including both text and figures) Shapley is misspelled as “Shapely”. Another example is in section 6 where the C in CNN is written as “convectional” rather than convolutional.
- Some terms need to be defined (or removed if not important). For example, in the large paragraph on page 2, the term “missingness” and “domain” are not very clear; I can guess what you mean but it would be better if it were clear. Likewise when describing the “appealing properties” of the Shapley value as a metric, it’s not clear yet what task you are hoping to use Shapley values for, and therefore not clear why these properties are appealing. Another instance of a similar lack of context/clarity is in the third contribution bullet point, which uses the words “all” and “both” without explaining what these are referring to. Another example is some acronyms; I know ERM but it should still be defined, as should RSC and IRM.
- In equation 2, do you use the same random subset m in all the experiments, or is it chosen randomly each time?
- Around equations 3 and 4, it would be good to explicitly define u and v as spatial coordinates and m and n as frequency coordinates.
- It would be good to give some more description of the datasets that were used in the experiments, particularly how many train and test images, how large each image is, and how many classes there are.
- Is the “modified image” in Figure 6 an actual result of the inverse Fourier Transform, or just an illustration? It looks like just the edges of the input image, which I doubt would appear naturally as a result of the CFA method, but it would be good to specify if this is just an illustration (or even better to show an actual image resulting from the method).

---

> ### Author Response · Authors · 2023-11-20
> **Thank you for providing us with valuable advice on writing. We have addressed every Weakness and Question in order to resolve your concerns.**
>
> *W1:*
>
> Thank you very much for your valuable suggestion about writing. We will make revisions and improve the paper further.
>
> *W2:*
>
> Via frequency-based analysis of two shifts,  we empirically explain why the performance of some algorithms deteriorates rapidly in OoD environments and why some algorithms outperform others in one type of domain shift while fail in another one. Based on these analysis, we further propose our CFA.
>
> By comparing the performance of different methods on two shifts, we can highlight the advantages of our CFA.
>
>
> *W3:*
>
> Shapley values of different frequency components is stored as two-dimensional matrix, as it can be seen from the  heatmap of Figure 4. After the Fourier transform, the center point is the zero frequency component, and the frequency gradually increases from the center point to all directions. In order to process this two-dimensional information into one-dimensional information, we sum the two-dimensional shapley values ringly and each ring represents a frequency point.
>
> The added and subtracted frequency components are derived from an average over training images and CFA is only used in the training process only. CFA helps the network learn more domain-generalizable features in the training stage. In testing stage, we use original testing images for fair comparison.
>
> *W4:*
>
> To make a strict ablation study, we run each experiment once with fixed random seed. In Table 2, we rerun each experiment with random seed for three times independently. We report the average of three runs, and its corresponding standard error.
>
> *W5:*
>
> The proof of nonlinearity is too complex to achieve, and the linear model can partially prove CFA's effectiveness.
>
> *W6:*
>
> Thank You for the reminder. $F$ means Discrete Fourier Transform. $f$ means the output of model. For classification task, it means the probabilities.
>
> *Q1 & Q2:*
>
> Sorry for the confusion, we will improve the writing further.
>
> *Q3:*
>
> The main purpose of listing the properties of the Shapley value here is to demonstrate that it is reasonable to use the Shapley  value to analyze the importance of different frequency components. The properties of the Shapley value has been proved in work[1].
>
> Missingness: The Shapley value of  player i is zero, if ∀S ⊆ N, V (S ∪ {i}) = V (S).
> [1] L. S. Shapley. A value for n-person games. In Contributions to the Theory of Games, page
> 307–317, 1953.
>
> In the third contribution bullet point, we will rephrase like "We conduct experiments to verify the effectiveness of our method on five baseline OoD algorithms on both diversity shift and correlation shift."
>
> As for the acronyms, ERM first appears in the abstract and we have defined for ERM. For other acronyms, we will give the definition for clarity.
>
> *Q4: In equation 2, do you use the same random subset m in all the experiments, or is it chosen randomly each time?*
>
> It is chosen randomly each time.
>
> *Q5: Around equations 3 and 4, it would be good to explicitly define u and v as spatial coordinates and m and n as frequency coordinates.*
>
> Thanks for your advice. It's better to give more explaination of the equations.
>
> *Q6: It would be good to give some more description of the datasets that were used in the experiments, particularly how many train and test images, how large each image is, and how many classes there are.*
>
> Our implementation of experiments is based on the OoD-bench and DomainBed suits, which are open-source and details like image resolution, training setup for different datasets are fixed. Thus, we do not give more details in the paper.  We  will provide more experiment details in the supplementary.
>
> [1] Nanyang Ye, Kaican Li, Haoyue Bai, Runpeng Yu, Lanqing Hong, Fengwei Zhou, Zhenguo Li,
> and Jun Zhu. Ood-bench: Quantifying and understanding two dimensions of out-of-distribution
> generalization, 2022.
>
> [2] Ishaan Gulrajani and David Lopez-Paz. In search of lost domain generalization. In International Conference on Learning Representations, 2021.
>
> https://github.com/facebookresearch/DomainBed
>
> *Q7: Is the “modified image” in Figure 6 an actual result of the inverse Fourier Transform, or just an illustration? It looks like just the edges of the input image, which I doubt would appear naturally as a result of the CFA method, but it would be good to specify if this is just an illustration (or even better to show an actual image resulting from the method).*
>
> The “modified image” is an illustration, which we use to emphasize our CFA pay more attention to the structure information. As in the visualization of modified image of its positive frequency bands, we can find more shape or structure information of the objects, which are more domain-invariant features.

---

> > ### Comment · Reviewer_BXsg · 2023-11-22
> >
> > Thanks to the authors for responding to my questions. Most of my concerns are addressed by these explanations, and I have raised my score accordingly. I did not raise my score further because as yet (as far as I can tell) these explanations and clarifications have not been implemented in the paper. I would also encourage the authors to always clarify in figure captions whenever a portion of a figure is using real data/model output versus a synthetic illustration, to avoid confusing future readers.

---

> > > ### Author Response · Authors · 2023-11-23
> > >
> > > Thanks for your suggestion. We have updated our paper online and we will keep to improve our writing!

---

### Official Review · Reviewer_4fGE · 2023-10-31

**Soundness:** 3 good
**Presentation:** 3 good
**Contribution:** 3 good
**Rating:** 6
**Confidence:** 3

**Summary:**

In this paper, authors proposed a novel method for OOD generalization based on augmentation in the frequency domains.
The results shown achieve state of the art performance in both diversity and correlation shifts.

**Strengths:**

- The proposed method shows consistent results in both diversity and correlation shifts. This is a relevant point as many OOD generalization methods usually achieve good results in only one of the two shifts

- The proposed method can improve the performance of existing algorithms for OOD, making it general and applicable in many contexts

- The explanation is clear

**Weaknesses:**

- I have some doubts about the experimental results, it seems that results are reported (e.g. for Colored MNIST) only for a certain degree of correlation/ratio between color and digit. How does the proposed method behave in under different degrees of correlation (as in [1])

- Related to the point above, the explanation about the nature of the shift in the different dataset is lacking. E.g. for MNIST, what ratio/correlation was used? For CelebA which attributes were considered? Etc.

- Comparison or references to relevant work in the debiasing/generalization fields are missing; e.g. [2,3,4]

[1] Lee, Jungsoo, et al. "Learning debiased representation via disentangled feature augmentation." Advances in Neural Information Processing Systems 34 (2021): 25123-25133.

[2] Nam, Junhyun, et al. "Learning from failure: De-biasing classifier from biased classifier." Advances in Neural Information Processing Systems 33 (2020): 20673-20684.

[3] Barbano, C. A., Dufumier, B., Tartaglione, E., Grangetto, M., & Gori, P. (2022). Unbiased supervised contrastive learning.The Eleventh International Conference on Learning Representations (ICLR), 2023.

[4] Lee, J., Park, J., Kim, D., Lee, J., Choi, E., & Choo, J. (2023, June). Revisiting the importance of amplifying bias for debiasing. In Proceedings of the AAAI Conference on Artificial Intelligence (Vol. 37, No. 12, pp. 14974-14981).

**Questions:**

See weaknesses;

Additional questions:

- I think that your method could provide some sort of "explanability" in terms of visual interpretation of certain frequencies. Could you add some examples e.g. for ColoredMNIST or CelebA showing the reconstructed images for the most important frequency (both positive and negative)?

- How where the $\alpha$ and $\beta$ hyperparameters chosen? How robust is your method to changes in these values?

- I suggest authors change the line "We introduce Shapley value" in the introduction as it seems to suggest that this paper proposes Shapley value rather then their novel application

---

> ### Author Response · Authors · 2023-11-20
> **We appreciate your constructive feedback. We have addressed every Weakness and Question in order to resolve your concerns.**
>
> *W1: I have some doubts about the experimental results, it seems that results are reported (e.g. for Colored MNIST) only for a certain degree of correlation/ratio between color and digit. How does the proposed method behave in under different degrees of correlation (as in [1])*
>
> Colored MNIST is transforming from [1] as follows: [1] defines three environments (two training, one test) ; first, assign a preliminary binary label y˜ to the image based on the digit: y˜ = 0 for digits 0-4 and y˜ = 1 for 5-9. Second, obtain the final label y by flipping y˜ with probability 0.25. Third, sample the color id z by flipping y with probability p, where p is 0.2 in the first environment, 0.1 in the second, and 0.9 in the test one. Finally, color the image red if z = 1 or green if z = 0.
>
> [1] Martin Arjovsky, Léon Bottou, Ishaan Gulrajani, and David Lopez-Paz. Invariant risk minimization, 2020.
>
> *W2: Related to the point above, the explanation about the nature of the shift in the different dataset is lacking. E.g. for MNIST, what ratio/correlation was used? For CelebA which attributes were considered? Etc.*
>
> According to [2], the OoD distribution shift can be divided into two patterns: diversity shift and correlation shift, referring to different reasons for domain shift. Diversity shift is caused by the appearance of new unseen environment features in the test set while
> correlation shift delineates the spurious correlation between environment features and labels.
>
> For CelebA, [2 ]treats “hair color” as the classification target and “gender” as the spurious attribute.
>
> For NICO,  consisting of real-world photos of animals and vehicles captured in a wide range of contexts such as “in
> water”, “on snow” and “flying”. There are 9 or 10 different contexts for each class of animal and vehicle. This dataset simulates a scenario where animals and vehicles are spuriously correlated with different contexts.
>
> [2] Nanyang Ye, Kaican Li, Haoyue Bai, Runpeng Yu, Lanqing Hong, Fengwei Zhou, Zhenguo Li, and Jun Zhu. Ood-bench: Quantifying and understanding two dimensions of out-of-distribution generalization, 2022.
>
> *W3: Comparison or references to relevant work in the debiasing/generalization fields are missing; e.g. [2,3,4]*
>
> We are reproducing LfF + BE  proposed in [4] on CMNIST used in our paper for fair comparsion. But we find that the CMNIST used in OoD generalization fields is quiet different from that in debiasing fields. In OoD generalization, in our CFA, we only use  object category as supervised information without extra arribute information as supervision. However, in [4], it seems that extra arribute information is needed in the training.
>
> *Q1: I think that your method could provide some sort of "explanability" in terms of visual interpretation of certain frequencies. Could you add some examples e.g. for ColoredMNIST or CelebA showing the reconstructed images for the most important frequency (both positive and negative)?*
>
> Thanks for your advice. In Figure 4, we reconstructed images of PACS. As it can be seen, the most important positive frequency is about shape or contour of the object, while the negative one is about the color of the object.
>
> *Q2: How the hyperparameters $\alpha$ and $\beta$  chosen? How robust is your method to changes in these values?*
>
> In the supplementary, we have evaluated the parameter sensitivity of CFA by changing one parameter and fixing the other. U-shaped parameter sensitivity curves are shown in Figure 9. From this figure, we can observe that suitable values of $\alpha$ and $\beta$ help to improve the generalization performances.
>
> *Q3: I suggest authors change the line "We introduce Shapley value" in the introduction as it seems to suggest that this paper proposes Shapley value rather then their novel application.*
>
> Thanks for your advice. To better and clear express, we rephrase this sentence as "We explore Shapley value".

---

> > ### Comment · Reviewer_4fGE · 2023-11-21
> >
> > I thank the authors for their response and for their clarifications. I am happy to confirm my score

---

### Official Review · Reviewer_L2GP · 2023-11-04

**Soundness:** 1 poor
**Presentation:** 2 fair
**Contribution:** 2 fair
**Rating:** 5
**Confidence:** 4

**Summary:**

The work analyses the importance of different frequency components to OOD generalization. The authors utilized Shapley values, which provide evidence of whether a certain frequency is favourable/unfavourable to generalization. Based on the analysis, they proposed a frequency augmentation technique, which benefits the OOD generalization of models.

**Strengths:**

+ Interesting idea to use Shapley values to analyse different algorithms (ERM, IRM,RSC) from a frequency perspective

**Weaknesses:**

__Lacks novelty: analysis and Shapley value__
The analysis approach has quite strong similarity to that of [1], but the authors did not explore or comment about any differences or improvements compared to previous work. Conceptually, the analysis is also similar to [5], which is not discussed.
Furthermore, there exist limitations on the approximation of Shapley values through random sampling of permutation proposed by Castro et al. [2], and Castro et al. further improved the approximation in [3], which was not applied by the authors.
There are some other weaknesses using Shapley-value based methods to explain feature importance (as discussed in [4]), which were not considered by the authors. For instance, different frequency components might be interrelated, but using permutation-based approach might not consider this correlation.


__Augmentation method__
The calculation of Shapley values is model-based, but it is unclear in either Algorithm 1 or section 4.1 whether the authors use a pre-trained model or the model under training.

__Experiment design__
- Experiments are limited to comparison with empirical risk minimization, invariant risk minimization, etc., while the proposed augmentation approach is not analyzed (and put in context with related works) in comparison with existing  state-of-the-art (frequency) augmentation approaches.
- Vague experiment details, e.g. unknown portion of the randomly sampled permutations, image resolution, training setup, unclear classification tasks on datasets like CelebA.
- Results are limited to MLP and ResNet18 models, without exploring Transformers or even other CNN architectures, and on small datasets. Generalization from ImageNet to e.g. ImageNet-R, ImageNet-O etc. should be studied.
- Formulas have unclear components or use of symboles:
	- ‘m’ was used twice in the equations (2) and (3)
	- no explanation for the designated set ‘T’ and the function f(.) in equation (6)
	- no explanation for ‘N’ when introducing the specific permutation π ∈ Π(N)

[1] Chen et al., “Rethinking and Improving Robustness of Convolutional Neural Networks: a Shapley Value-based Approach in Frequency Domain” (2022)

[2] Castro et al.,  “Polynomial calculation of the Shapley value based on sampling” (2009)

[3] Castro et al.,  ”Improving polynomial estimation of the Shapley value by stratified random sampling with optimum allocation” (2017)

[4] Kumar et al. “Problems with Shapley-value-based explanations as feature importance measures” (2020)

[5] Wang et al. "What do neural networks learn in image classification? A frequency shortcut perspective", (2023)

**Questions:**

- How is the analysis approach different from [1], which analyses the importance of frequency components to adversarial robustness, except that the authors extend it to OOD generalization? Also, what are the relations with [5]?
- In the introduction, the authors claim that the augmentation method is model-agnostic. But in sec. 3.1, they claim that the calculation of Shapley values is based on model output. These statements are contradictory to each other. Can the authors clarify the inconsistency?
- The proposed augmentation approach is class-wise and from the matrix of Shapley values, I infer the image resolution is low. Do the experiments show the feasibility of CFA to datasets with thousands classes (e.g. ImageNet1K)? How does the resolution of images affect the calculation of Shapley value?
- To approximate the Shapley values, the authors randomly sample a portion of the permutation? What is the value of ‘m’, the portion, and how does this affect the calculation of stable Shapley values? Is there any trade-off between efficiency and stability?
- The authors mentioned the OOD generalization for deep neural networks, but the experiments only show results for MLP and ResNet18. What about other CNNs and transformers?

---

> ### Author Response · Authors · 2023-11-20
> **We appreciate your constructive feedback. We have addressed every Weakness and Question in order to resolve your concerns.**
>
> *W1:*
>
> [1] uses Shapley values to analyse the importance of frequency components to adversarial robustness, while we extend Shapley values to OoD generalization. [1] and our method focus on different research areas, but we both resort to Shapley values to analyse in frequency domain.
>
> [5] finds that neural networks can exhibit different frequency biases, tending to adopt frequency shortcuts based on data characteristics because of simplicity-bias learning.They foresee that avoiding frequency shortcut learning may hold promise in improving generalization. But they do not give a concrete method to improve generalization yet. In our method, we precisely attribute which frequency components of training images are important for generalization via Shapley values and propose CFA to improve generalization.
>
> *W2:*
>
> Thanks for providing us more profound methods of the approximation of Shapley values. Intuitively, if we can attribute which frequency components of training images are important for generalization more precisely, CFA can have better performance in OoD generalization. We will try to use the improved approximation of Shapley values in [3] to refine our CFA.
>
> *W3:*
>
> In our method, the first step is to calculate Shapley values of existing Ood algorithms like ERM,IRM. Then, based on calculated Shapley values, we carry out CFA to improve corresponding OoD algorithms.
>
> *W4:*
>
> In Table 2, DFF proposed by Lin et al. (new SOTA in CVPR, 2023) is to modulate the frequency components to learn domain-generalizable features, which is the most similar one to our CFA. Thus, we mainly compare our CFA with DFF cross seven datasets.
>
> *W5:*
>
> Our implementation of experiments is based on the OoD-bench and DomainBed suits, which are open-source and details like image resolution, training setup for different datasets are fixed. Thus, we do not give more details in the paper. For readers nor familar with OoD-bench or DomainBed, we will provide more experiment details in the supplementary.
>
> [1] Ishaan Gulrajani and David Lopez-Paz. In search of lost domain generalization. In International Conference on Learning Representations, 2021.
>
> https://github.com/facebookresearch/DomainBed
>
> As for the details of calculation of Shapley values, we should supplement further in the paper. For image resolution of 224, we divide the image to $16 \times 16$ patches in the frequency domain and sample 1000 times for each sample.
>
> *W6:*
>
> We implement more experiments with Vit on two datasets. As it can be seen, CFA stills have impressive performance. Experiments on other datasets are still running.
> | Backbone | Dataset | Algorithm | Baseline | S    | W    | S+W                              |
> |----------|---------|-----------|----------|------|------|----------------------------------|
> | Vit      | CelebA  | ERM       | 82.0     | 84.4 | 83.7 | 85.0 ($\uparrow$3.0）  |
> |          | PACS    | ERM       | 82.4     | 86.0 | 85.8 | 86.7  ($\uparrow$4.3) |
>
> As for large-scale datasets, we alreary experiment on DomainNet, which is a dataset of common objects in six different domains and 345 categories (classes). Domainet has around 600,000 images, while ImageNet-R has 200 ImageNet classes resulting in 30,000 images. Thus, we believe that experiments on DomainNet is quite convincing.
>
> *W7:*
>
> We randomly sample m=1000 of all permutations  for each data sample.  N means  all the permutations of image patches, while T means  ramdomly remain some frequency components to calculate Shapley values.
>
> *Q1 is similar to W1, please see the response to W1.*
>
> *Q2:*
>
> "Model-agnostic" is used to describe a class of algorithms that do not make any assumptions about the underlying model. The model-agnostic algorithms view the model as a black box and focus on understanding the relationship between the input and the output of model. Our CFA can be directly adopted to any model without modifying the structure of the network. In addition, the calculation of Shapley values in our paper also meets the definition of "model-agnostic", as it focuses on understanding the relationship between the input and the output of model without attention to the internal model.
>
> *Q3:*
>
> In OoD algorithms, 224 is a common-used resolution. For fair comparisons with other OoD lgorithms, we do not experiment on different resolutions.
>
> *Q4:*
>
> According to Castro et al [2], with the increasing of sampling times m, the average error for the estimation of Shapley value is decreasing. But considering the time complexity analysis below, we set m=1000 to better balance efficiency and stability.
>
> | m | 1000 | 10000 |
> |-------------|---------------|---------------|
> | e_ave     | 0.000684  | 0.000252  |
> |   time consumed(s)    | 1410.82    | 14519.82 |
>
> *Q5 is similar to W6, please see the response to W6.*

---

### Meta-Review · Area_Chair_g8Xs · 2023-12-07

**Metareview:**

This paper studies OOD generalization in the frequency domain. It proposes Class-wise Frequency Augmentation (CFA) to augment favorable frequency components and inhibit unfavorable ones.  The reviewers have raised a number of questions about the method, evaluation and explanation.  Some of those questions were answered during author-reviewer discussions. However, the reviewers do not consider the work publishable in its current form.  Here are two post-discussion notes from the reviewers: “It is promising, but I would not accept it in its present form”; “I did not raise my score further because as yet (as far as I can tell) these explanations and clarifications have not been implemented in the paper.”

**Justification For Why Not Higher Score:**

See above

**Justification For Why Not Lower Score:**

See above

---

### Decision · Program_Chairs · 2024-01-16

Reject